# Shorter Grazing Time and Supplementation Are Beneficial for Gastrointestinal Tract Development and Carcass Traits of Growing Lambs

**DOI:** 10.3390/ani12070878

**Published:** 2022-03-30

**Authors:** Yanmei Jin, Muhammad Asad, Xiaoqing Zhang, Jize Zhang, Ruizhi Shi

**Affiliations:** 1Marine College, Shandong University, Weihai 264209, China; jinym2001@sohu.com; 2Agrobiology and Plant Stress Physiology Lab., Department of Agronomy, University of Agriculture Faisalabad, Faisalabad 38000, Pakistan; ch.muhammadasad007@gmail.com; 3Institute of Grassland Research, Chinese Academy of Agricultural Sciences, Hohhot 010010, China; 4Institute of Practaculture Science, Tibet Academy of Agricultural and Animal Husbandry Sciences, Lhasa 850000, China; r231119569@163.com

**Keywords:** grazing schedule, gastrointestinal tract, muscle fiber, carcass quality

## Abstract

**Simple Summary:**

A shorter grazing time and supplementation have been demonstrated to increase the grazing efficiency and growth performance of lambs. However, the effects of a shorter grazing time on the gastrointestinal development and carcass quality of growing lambs are poorly understood. In the present study, lambs were offered 2, 4, 8 or 12 h of grazing and separately fed supplementation when off pasture. The results showed that, in comparison to longer grazing times, shorter grazing times—especially 4 h of grazing per day with supplementation—best improved gastrointestinal tract and carcass quality. Therefore, restricting the grazing of lambs to 4 h per day instead of grazing for more extended periods is a better grazing management approach in Inner Mongolia.

**Abstract:**

The effects of restricted grazing durations on the gastrointestinal development and carcass quality of growing lambs are poorly understood. In this study, 32 lambs were randomly assigned to four groups (*n* = 8, body weight = 21.86 kg) corresponding to 2, 4, 8 and 12 h of grazing per day. When off-pasture, all lambs were housed and fed concentrate and hay. When the grazing time decreased from 12 h to 2 h, the abomasum weight and large intestine length decreased (*p* = 0.019; *p* = 0.069). Compared to lambs grazed for 12 h, animals grazed for 2–4 h had a greater villus height and villus-to-crypt ratio in the duodenum, jejunum and ileum segments (*p* < 0.05); the 2 h lambs had superior carcass quality and a smaller diameter and area of the *gluteus medium* muscle fibers (*p* < 0.05), with no significant change after 4 h of grazing. The results indicated that shorter grazing times and supplementation were beneficial for the gastrointestinal tract development and carcass quality of growing lambs. Therefore, a better grazing management approach in Inner Mongolia could be to restrict the grazing of lambs to 4 h per day instead of grazing for more extended periods.

## 1. Introduction

Grazing management should be a win–win situation for both pasture and animals. However, the pursuit of higher animal production by traditional grazing management approaches has led to considerable overgrazing and the unsustainable utilization of grassland in Inner Mongolia, China. In local, traditional grazing systems, animals spend considerable time grazing, which exacerbates trampling damage to the grassland and increases the energy demand for maintenance instead of growth [1]. On the other hand, the established confinement feeding system relieves grassland pressure and improves the potential of animal production but it also has multiple disadvantages, such as environmental pollution, increased farming costs and the impact of fatty acid metabolites on animal health [2,3]. Therefore, to successfully develop both the grassland ecosystem and high-quality livestock products, restricted grazing with supplementation is attracting increasing attention.

Restricted grazing may derive from the time-controlled grazing proposed by Hart et al. [4]. Reduced grazing duration requires a smaller workforce, giving herdsmen more time for other jobs. Concurrently, restricted time at pasture with supplementation improves the growth performance of grazing sheep [5]. Our previous study found that restricting the grazing of lambs to 4 h per day improved grazing efficiency [6] and increased health-promoting *n*-3 unsaturated fatty acids in muscle tissue [7]. Restricted grazing with supplementation reduced the amount of movement and pasture intake of grazing sheep [6,7], which affected the development of the gastrointestinal tract as well as muscle fiber and carcass quality. The development of the gastrointestinal tract plays a vital role in nutrient digestion and absorption and muscle fibers impact meat quality [8]. However, the duration of restricted grazing time that has the most beneficial effect on the critical physical indices that are closely related to gastrointestinal tract development and carcass quality in growing lambs is still unknown.

Accordingly, this study explored the effects of restricted-time grazing with supplementation on the morphological characteristics of the gastrointestinal tract and muscle fibers as well as the carcass quality of lambs. It was hypothesized that shorter grazing duration and supplementation would improve the gastrointestinal development and carcass quality of growing lambs. The optimum grazing duration would be beneficial for both animal production and the sustainable utilization of grassland, thereby providing a helpful reference for developing a suitable grazing management system in Inner Mongolia.

## 2. Materials and Methods

### 2.1. Animals and Experimental Design

The present study was conducted according to the guidelines of the Office of Beijing Veterinarians (The Agriculture Ministry, Beijing, China). The Committee on Experimental Animal Management of the Chinese Academy of Agricultural Sciences (Beijing) approved the research protocol (Approval No. 35/16.03.2011).

The experiment was conducted over 99 d in the grazing season (from early July to late September) on an experimental farm located in the Xilin River Basin in Inner Mongolia (116°30′ E, 44°49′ N; alt. 1200 m ASL). The natural conditions of the study area as well as the quantity and quality of the pasture were reported previously [6]. A total of 32 weaned and castrated Ujumuqin lambs of similar body weight (21.86 ± 0.38 kg) and age (120 ± 15 days) were randomly allocated to one of four groups (*n* = 8): 2 h access to pasture (2H), 4 h access to pasture (4H), 8 h access to pasture (8H), and 12 h access to pasture (12H; control). When off-pasture, all lambs were housed and fed concentrate and grass hay supplements. The details regarding the experimental design, concentrate and grass hay supplementation, herbage allowance, individual feed and nutrient intakes and calculation of pasture dry matter (DM) intake for each treatment were described previously [6,7] and are listed in Appendix A.

Lambs had ad libitum access to water and salt blocks during the entire experimental period. All animals were slaughtered on the last day of the experiment. Before slaughter, lambs were fasted for 24 h and weighed to obtain slaughter weight values.

### 2.2. Sample Collection

To evaluate the apparent digestibility of DM, fecal samples were collected using fecal bags as detailed previously [6]. In the slaughter experiment, the carcasses of lambs were weighed. The empty weights of digestive organs (rumen, reticulum, omasum, abomasum and intestine) and the lengths of the small and large intestines were recorded. The backfat thickness of the *longissimus dorsi* (LD) muscle was determined using procedures described in previous research [9]. Then, muscle samples (3–5 g per sample) were cut from the center of the LD and *gluteus medius* (GM) muscles of the lamb carcass and fixed in 10% (*w*/*v*) formalin in phosphate buffer (pH 7.4) for further histological analysis. Samples of the duodenum, jejunum and ileum were collected as described in Wang et al. [10] and fixed in 10% formalin buffer as described above for future morphological analysis.

### 2.3. Chemical Analysis

Feed and fecal samples of DM were analyzed according to the method of the Association of Official Analytical Chemists (AOAC; method 934.01, 1990) [11].

### 2.4. Morphometric Analysis

All histological sections of muscle and intestine samples were photographed using an optical microscope equipped with an ocular micrometer (SMZ745T; Nikon Microscope Co., Tokyo, Japan). Ten photomicrographs of each muscle sample section were obtained at 10 × 40 magnification. Image-Pro Plus (IPP, version 6.0; Media Cybernetics, Silver Spring, MD, USA) was used to correct the scale and measure the number, diameter and density of muscle cells in each photomicrograph. Approximately 200 fibers of each sample were counted to evaluate the muscle fiber features. The villus height and crypt depth of well-oriented crypt–villus units (*n* = 10 per photomicrograph) in different intestinal segments were estimated using the same approach as for muscle fibers, using IPP. The villus height and crypt depth were measured as the mean distance from the tip to the base of the villus and the base of the crypt to the crypt–villus junction, respectively.

### 2.5. Statistical Analysis

The small intestine and muscle fiber morphometric data were analyzed using the ANOVA procedure in SAS (version 8.2) as a completely randomized design with a model that included treatment effects and experimental error. Each individual animal was considered an experimental unit. The DM digestibility data from each treatment in July, October and September were analyzed using repeated measures in the MIXED procedure of SAS. The following model was used:Yjki = μ + Ak + Bi + ABki + εjki 
where Yjki is the target variable, μ is the overall mean, Ak is the fixed effect of the treatment, Bi is the fixed effect of the month, ABki is the interaction effect of the treatment × month and εjki is the residual error. Differences among treatment means were compared using Tukey’s test. A probability (*p*) value of <0.05 was considered statistically significant and 0.05 ≤ *p* < 0.10 was considered as a tendency to differ.

## 3. Results

### 3.1. Gastrointestinal Tract Characteristics

The major digestive organs of lambs were not significantly affected by the restricted grazing treatments (Table 1). However, the abomasum weight of the 2H lambs was significantly lower than that of the 12H lambs (*p* = 0.019), and there were no significant differences among the 4H, 8H and 12H lambs. Additionally, the large intestine tended to be longer (*p* = 0.069) in the 8H and 12H groups compared to the 2H and 4H groups.

Pronounced differences in the morphological characteristics of the small intestine were observed among the four groups (Table 2). The duodenal villus height was greater in 2H lambs than in 8H and 12H lambs (*p* = 0.008) and there was no difference between the 2H and 4H groups. The crypt depth tended to be shallower (*p* = 0.076) in the 4H lambs compared to the 8H and 12H lambs. The villus-to-crypt ratio (V/C) in the duodenum segment was significantly decreased in the 8H and 12H groups compared to the 2H and 4H groups (*p* = 0.009), but there was no difference between the 2H and 4H groups. The villus height and V/C in the jejunum segment were greatest in the 2H and 4H lambs (*p* = 0.043; *p* = 0.039), with the lowest values observed in the 12H lambs, and there were no differences among the 2H, 4H and 8H lambs. Similarly, the villus height in the ileum segment and the V/C in the ileum segment were greater in the 2H, 4H and 8H lambs (*p* = 0.049; *p* = 0.014) compared to in the 12H lambs.

### 3.2. DM Digestibility

From July to September, lambs in the 12H group had the lowest average DM digestibility compared to other lambs (*p* < 0.05) and there were no significant differences among the 2H, 4H and 8H groups of lambs (Figure 1).

### 3.3. Morphological Evaluation of Muscle Fiber

As shown in Table 3, there were no significant differences in the morphological features of the LD muscle fibers among the four groups. However, the GM muscle fibers of the 2H group were smaller in diameter and area (*p* = 0.002; *p* = 0.038) and denser (*p* = 0.018) than in the 8H and 12H groups. Lambs in the 4H group had intermediate morphological values for the GM muscle and there were no significant differences between the 4H and 2H or 8H and 12H lambs.

### 3.4. Carcass Quality Traits

The main carcass quality traits of lambs were significantly affected by the restricted grazing treatment (Table 4). The 2H lambs had the highest carcass weight (15.9 kg; *p* = 0.039), the 12H lambs had the lowest value (13.8 kg), and the 4H lambs had an intermediate value. The dressing percentage was higher (*p* = 0.009) for the 2H, 4H and 8H lambs than for the 12H lambs. When the grazing time decreased from 12 to 2 h, the backfat thickness increased from 3.40 to 6.20 mm (*p* = 0.041). No significant difference was found between the 2H and 4H lambs.

## 4. Discussion

### 4.1. Effect of Restricted Grazing with Supplementation on the Gastrointestinal Tract

Ruminant animals possess a huge and complex gastrointestinal tract. A ruminant’s stomach has four compartments—rumen, reticulum, omasum and abomasum—in which the forestomach (rumen, reticulum, and omasum) plays a major role in the storage, fermentation, and decomposition of cellulose. The abomasum is the true stomach and has an actual digestive role [12]. In the present study, there were no significant differences in the forestomach weights of lambs among the four groups. However, lambs restricted to 2 h of grazing had a lower abomasum weight, which was attributed to their higher, more concentrated intake. Wang et al. [13] reported that the mucosal and muscular layers of the abomasum in Small-tailed Han sheep decreased following confinement treatment compared with grazing treatment, as the low-fiber feed passed through the digestive tract relatively easily due to the reduced volume of chyme. It was assumed that extended grazing enhanced the digestive function of the abomasum of sheep due to the stimulation provided by increased forage intake, so the abomasum was greater in the lambs that grazed for 12 h compared to 2 h. 

As an essential absorptive site of nutrients in animals, the small intestine can be manipulated by feeding strategies. In this study, although the weight and length of the small intestine of lambs were unchanged across the different groups, the morphological characteristics of the small intestine were influenced by the different durations of the restricted grazing conditions. Lambs that grazed for 2 or 4 h and received more concentrated supplementation had a greater villus height and V/C of the duodenum, jejunum and ileum compared to lambs that grazed for 12 h and received less concentrated supplementation. This result was consistent with a previous study in which goat kids that received concentrated supplementation had increased villus height in the jejunum and ileum compared to grazing kids [14]. The morphologies of intestinal villi and crypts are some of the most important indicators of the digestive and absorptive capacity of the small intestine. An increased villus height and V/C indicates an enhancement of digestive and absorption functions. It is well known that dietary factors, including nutrient levels and the composition and type of diet, impact the intestinal mucosal structure and its digestive and absorptive capacities [15,16]. High-forage diets decrease diet digestibility [17] and nutrient flow in the small intestine, inducing loss of the villi epithelium [18]; high levels of diet digestibility are associated with enlargement of the intestinal villi and an increased V/C [19]. Under similar nutritional levels across the four groups, the forage intake from pasture gradually increased as the grazing time was extended, which resulted in a lower villus height and V/C for the lambs that grazed for 12 h compared to 2–4 h. This indicated that grazing for longer than 4 h and increased forage intake had an unfavorable effect on the digestion and absorption functions of the small intestine.

Restricted grazing also affects the development of the large intestine. In the present study, the large intestine of lambs that grazed for 8–12 h tended to increase in length. The large intestine provides prolonged storage, intensive mixing, and slow aboral transfer of the digesta. A previous study found that the length of the large intestine was related to the feeding type [20]. When the ratio of forage to concentrate in the diet increased, the nutrient flow to the large intestine of sheep also increased by decreasing ruminal residence time and digestion [21]. Accordingly, it was deduced that extending the grazing time for sheep increases the forage intake, which necessitates a longer large intestine and longer retention time for digesta in the large intestine for increased hindgut fermentation.

In summary, shorter grazing durations are beneficial to the gastrointestinal tract development of lambs as these result in decreased foraging and more concentrated intake, which, in turn, result in higher nutrient digestibility.

### 4.2. Effect of Restricted Grazing with Supplementation on the Morphological Features of Muscle Fiber

Skeletal muscle is the largest tissue of locomotion in the body [22] and increased locomotion affects corresponding muscle fiber features [23]. In this study, an increase in the diameter and area and a decrease in the density of GM muscle fibers were observed as grazing duration increased. However, there was no difference in the morphological characteristics of LD muscle fibers across treatments. This was probably due to the different locations and locomotion patterns of the GM and LD muscles in the body. Multiple studies have indicated that physical training or spontaneous activity is positively correlated with the mean fiber cross-sectional area or relative area. This is especially true for GM muscle fibers, which exhibit the potential to adapt metabolically and structurally as a result of training [24,25]. Our previous study [6] (using the same animals and the same grazing schedule) revealed that lambs that were restricted to 2–4 h of grazing covered 43–53% less distance (2.2–3.3 vs. 4.9–5.6 km/day) and walked for less time (11.8–17.5 vs. 25.9–28.4 min per grazing period) compared to lambs that grazed for 8–12 h. So, the differences in GM muscle fiber structure between these lambs were primarily due to differential walking locomotion during grazing. Unlike the GM muscle, the LD muscle is highly activated as an extensor or flexor of the vertebral column under galloping or jumping conditions; in walking and trotting—which are the main locomotion patterns during grazing—it is less activated [25]. 

Additionally, variations in the morphological characteristics of muscle fibers, such as their diameter and cross-sectional area, influence meat quality. As Maltin et al. (1998) [8] stated, muscles with larger fibers result in meat with less tenderness and greater toughness than meat from muscles with smaller fibers. Therefore, the restriction of grazing to 2–4 h per day has the potential to improve meat quality by improving the morphological characteristics of the muscle fibers of grazing animals. 

### 4.3. Effect of Restricted Grazing with Supplementation on Carcass Quality

As mentioned above, lambs that grazed for a shorter time and received more concentrated supplementation had higher DM digestibility and greater absorption of nutrients from the gut, which resulted in higher slaughter performance, especially carcass weight and backfat thickness. Studies have generally established that the supplementation of grazing animals with concentrates containing protein, energy or a protein–energy combination promotes DM digestibility, forage use efficiency and even weight-related carcass traits due to a better balance between energy and nitrogen [26,27]. On the other hand, the study by Zhang et al. [6] demonstrated that lambs that grazed for 2 or 4 h showed a remarkable decrease in extra energy expenditure through a significant reduction in their grazing activity; this, in turn, increased their available energy and resulted in a positive energy balance, thereby improving animal production. Overall, for all the reasons described above, grazing for shorter periods was beneficial to animal production.

## 5. Conclusions

This study suggests that, compared with longer grazing durations, shorter grazing durations—especially restrictions to 4 h of grazing per day with supplementation—promote gastrointestinal tract development, optimize muscle fiber characteristics and consequently improve the carcass quality of growing lambs. Therefore, in typical pastoral livestock systems in Inner Mongolia, there is the potential for better grazing management to be achieved by restricting the grazing of lambs to 4 h per day instead of grazing for more extended periods.

## Figures and Tables

**Figure 1 animals-12-00878-f001:**
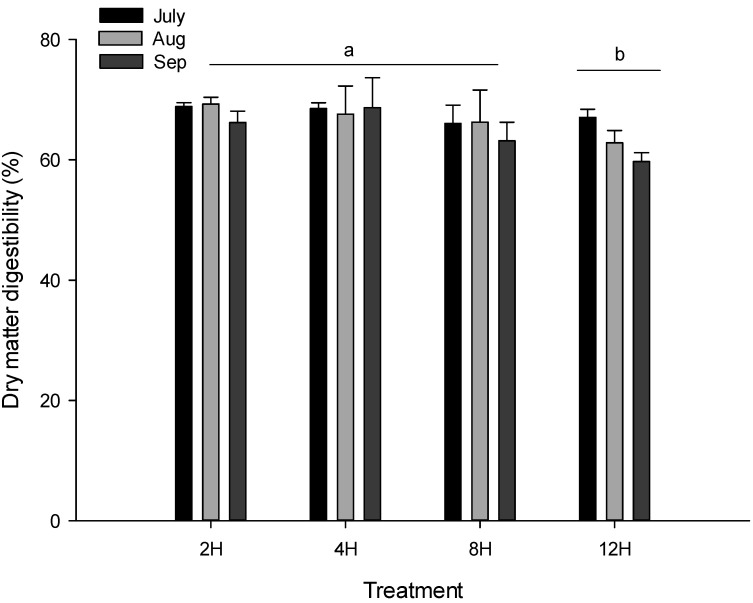
Dry matter digestibility by lambs in each treatment group. Treatments: 2H = 2 h access to pasture; 4H = 4 h access to pasture; 8H = 8 h access to pasture; 12H = 12 h access to pasture. The ^a^ and ^b^ mean there was significant difference (*p* < 0.05) among treatments.

**Table 1 animals-12-00878-t001:** Digestive organs of lambs following different grazing treatments.

Item	Treatment ^1^				SEM	*p*-Value
	2H	4H	8H	12H		
Rumen (g)	703.60	705.20	723.00	730.33	11.67	0.824
Reticulum (g)	103.00	104.00	107.00	108.40	3.04	0.348
Omasum (g)	81.60	77.60	78.40	80.80	3.60	0.980
Abomasum (g)	91.33 ^b^	105.20 ^ab^	104.00 ^ab^	107.00 ^a^	2.61	0.019
Small intestine (g)	530.00	544.33	507.67	523.33	15.8	0.885
Large intestine (g)	334.40	335.60	380.40	347.20	19.8	0.853
Small intestine length (m)	25.57	25.10	26.17	26.00	0.41	0.815
Large intestine length (m)	4.19	5.69	6.41	6.49	0.38	0.069

^1^ Treatments: 2H = 2 h access to pasture; 4H = 4 h access to pasture; 8H = 8 h access to pasture; 12H = 12 h access to pasture. SEM: standard error of the mean. Means within a row with different superscripts differ significantly (*p* < 0.05).

**Table 2 animals-12-00878-t002:** Morphological characteristics of the small intestine of lambs following different grazing treatments.

Item	Treatment ^1^				SEM	*p*-Value
	2H	4H	8H	12H		
Duodenum						
Villus height (µm)	564.91 ^a^	544.96 ^ab^	535.00 ^b^	506.91^c^	7.26	0.008
Crypt depth (µm)	353.38	346.58	370.28	369.10	4.11	0.076
V/C	1.60 ^a^	1.57 ^a^	1.45 ^b^	1.37 ^b^	0.03	0.009
Jejunum						
Villus height (µm)	662.75 ^a^	675.81 ^a^	657.22 ^ab^	622.76 ^b^	7.51	0.043
Crypt depth (µm)	425.33	423.91	434.56	442.89	4.16	0.375
V/C	1.57 ^a^	1.60 ^a^	1.51 ^ab^	1.41 ^b^	0.03	0.039
Ileum						
Villus height (µm)	645.25 ^a^	653.85 ^a^	651.02 ^a^	614.80 ^b^	6.03	0.049
Crypt depth (µm)	404.10	403.62	410.60	428.91	4.22	0.091
V/C	1.60 ^a^	1.62 ^a^	1.59 ^a^	1.43 ^b^	0.03	0.014

^1^ Treatments: 2H = 2 h access to pasture; 4H = 4 h access to pasture; 8H = 8 h access to pasture; 12H = 12 h access to pasture. SEM: standard error of the mean. Means within a row with different superscripts differ significantly (*p* < 0.05).

**Table 3 animals-12-00878-t003:** Morphological characteristics of *longissimus dorsi* and *gluteus medius* muscles of lambs following different grazing treatments.

Item	Treatment ^1^				SEM	*p*-Value
	2H	4H	8H	12H		
*Longissimus dorsi* muscle						
Diameter (µm)	26.94	28.60	29.95	28.86	0.49	0.200
Area (µm^2^)	651.26	655.96	701.14	705.54	24.50	0.517
Density of fibers (*n*/mm^2^)	1049.97	1001.64	984.77	980.28	18.71	0.461
*Gluteus medius* muscle						
Diameter (µm)	30.36 ^b^	33.72 ^ab^	34.35 ^a^	36.46 ^a^	0.72	0.002
Area (µm^2^)	783.85 ^b^	969.99 ^ab^	1011.80 ^a^	1083.69 ^a^	37.20	0.038
Density of fibers (*n*/mm^2^)	771.79 ^a^	688.40 ^ab^	641.35 ^b^	635.79 ^b^	20.57	0.018

^1^ Treatments: 2H = 2 h access to pasture; 4H = 4 h access to pasture; 8H = 8 h access to pasture; 12H = 12 h access to pasture. SEM: standard error of the mean. Means within a row with different superscripts differ significantly (*p* < 0.05).

**Table 4 animals-12-00878-t004:** Slaughter performance of lambs following different grazing treatments.

Item	Treatment ^1^				SEM	*p*-Value
	2H	4H	8H	12H		
Slaughter weight (kg)	33.07	32.40	31.30	30.65	0.49	0.311
Carcass weight (kg)	15.93 ^a^	15.27 ^ab^	14.83 ^ab^	13.80 ^b^	0.27	0.039
Dressing percentage (%)	48.17 ^a^	47.12 ^a^	47.41 ^a^	45.02 ^b^	0.37	0.009
Backfat thickness (mm)	6.20 ^a^	5.80 ^ab^	4.00 ^b^	3.40 ^b^	0.14	0.041
*Longissimus dorsi* muscle area (cm^2^)	18.41	17.56	17.15	17.13	0.45	0.759

^1^ Treatments: 2H = 2 h access to pasture; 4H = 4 h access to pasture; 8H = 8 h access to pasture; 12H = 12 h access to pasture. SEM: standard error of the mean. Means within a row with different superscripts differ significantly (*p* < 0.05).

## Data Availability

Data generated or analyzed during this study are included in this published article.

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
