# Peer review of "Shorter Grazing Time and Supplementation Are Beneficial for Gastrointestinal Tract Development and Carcass Traits of Growing Lambs"

_animals, 2022, doi:10.3390/ani12070878_

Round 1

Reviewer 1 Report

Reviewer(s)’ General Comments to Authors:

Minnor comments

Line 39: …approaches have led to…

Line 56: … muscle fibre and carcass…, fibre or fiber, which is more suitable?

Line 82:… each treatment was described…

Line 89: faecal? Maybe fecal?

Line 111: … crypt depth was measured…

Line 118: … and September was analysed using…

Line 192: … sheep was decreased…

Line 201: … intestine was influenced…

Line 273: …of grazing for more extended periods…

The study is interesting and the manuscript is well written and comprehensive. There are areas of the manuscript where english grammar could be improved. There are a few typos,  these are the type of mistakes that will not be caught by spell check, so please read through carefully. It is hoped that if the manuscript is considered for acceptance that through the efforts of the authors and the editorial staff, improvements can be made. It should be stated where or under which conditions (regions or countries). The authors have made a great effort, measuring many variables which provide a very nice picture of the digestive organs of lambs and morphological characteristics of the small intestine of lambs following different grazing treatments. It is suggested to include a hypothesis supporting the inclusion levels and expected results. Providing and testing hypothesis helps to increase nutritional knowledge. 

Author Response

Point 1: Line 39: …approaches have led to…

Response 1: Sincerely thanks for all you have done on this work. We asked a native speaker to read the whole paper and made some furtehr slight changes to wording/sentence structure where needed. But in sentence: However, the pursuit of higher animal production by traditional grazing management approaches has led to …; here the verb should use “has” in line 40.

Point 2: Line 56: … muscle fibre and carcass…, fibre or fiber, which is more suitable?

Response 2: In response to your comment, we changed all ‘fibre’ to ‘fiber’ in the revised MS. Additionally, We marked up some additional changes, keeping the USA spelling and punctuation.

Point 3: Line 82:… each treatment was described…

Response 3: The details … were …, so that should be “were” in line 88.

Point 4: Line 89: fecal? Maybe fecal?

Response 4: Changes have been made in lines 94-95 and 106.

Point 5: Line 111: … crypt depth was measured…

Response 5: The villus height and crypt depth were measured…, here should use “were” in line 118.

Point 6: Line 118: … and September was analysed using…

Response 6: The DM digestibility data from each treatment in July, October, and September were analyzed …, here should use “were” in line 126.

Point 7: Line 192: … sheep was decreased…

Response 8: The wording of were has been removed from line 204.

Point 8: Line 201: … intestine was influenced…

Response 8: …, the morphological characteristics of the small intestine were influenced …, here should use “were” in line 213.

Point 9: Line 273: …of grazing for more extended periods…

Response 9: Correction has been made in lines 21, 34 and 292.

Point 10: The study is interesting and the manuscript is well written and comprehensive. There are areas of the manuscript where english grammar could be improved. There are a few typos,  these are the type of mistakes that will not be caught by spell check, so please read through carefully. It is hoped that if the manuscript is considered for acceptance that through the efforts of the authors and the editorial staff, improvements can be made. It should be stated where or under which conditions (regions or countries). The authors have made a great effort, measuring many variables which provide a very nice picture of the digestive organs of lambs and morphological characteristics of the small intestine of lambs following different grazing treatments. It is suggested to include a hypothesis supporting the inclusion levels and expected results. Providing and testing hypothesis helps to increase nutritional knowledge.

Response 10: Thank you for your thorough comments. We have correct all the type of mistakes throughout text according to your suggustion.

For “It should be stated where or under which conditions (regions or countries)”: The region is in Inner Mongolia, which has been added in the “Abstract” (line 32) and “Conclusions” sections (line 290, page 8).

Additianlly, the hypothesis in lines 66–68 of the manuscript has been improved this time. That is: It was hypothesized that shorter grazing duration and supplementation will improve the gastrointestinal development and carcass quality of growing lambs.

Reviewer 2 Report

Shorter grazing time and supplementation are beneficial for gastrointestinal tract development and carcass traits of growing lambs

This manuscript is original and has a good novelty by providing insights regarding how the restriction of grazing hours can affect GIT development addressing villus characteristics. It is a well-written presentation and expands our knowledge with respect to sustainable agriculture and how it could be beneficial for grassland preservation not to be overgrazed and, at the same time, beneficial for the growth of lambs. The Abstract lacks some necessary information that I addressed in a detailed comment. The Introduction and hypothesis (objective) is presented well but need to hint at the economical value of grazing in the text. The results are clear and to the point and then well discussed. The conclusion at the end of the Abstract and in the separate section at the end of the manuscript is to the point and supported by the obtained results. The supplementary materials are enough informative, but authors should be consistent in decimal points (either one or two but consistent). Please edit the data accordingly.

My major concern is regarding M&M and Discussion. M&M has a lack of information regarding the access to water and feed and how the grazing management was performed. Please refer to my below sectional comment to address the lack in the text.

The authors should also speculate the obtained result by attributing the facts and mechanism that brought about these results in each sub-section rather than only providing evidence from other research.

I have some comments to improve the presentation of this work and I invite the authors to respond to my comments carefully and in the text.

I am sure this work can receive attention from the avid readers of Animals when published and thus give my highest recommendation to further processing of this work for publication after revision is completed.

Comments

Simple summary

It is all good.

Abstract

Please stat the number of animals in each group of grazing plus the ave. BW of the animals.

L25: please state what type of supplements are in the text.

Introduction

Please include a sentence (where appropriate) regarding the economic benefits of grazing that can add value to the indirect beneficial reasons behind doing this research.

M&M

L84: How the ad libitum water was provided during grazing? In buckets?

What was the acre of grazing for each group?

What happened during rain for housing the animals?

How was the grazing management performed?

Results

Please be consistent in using decimal points in the value of the tables.

Discussion

In each section, please improve the discussion by speculating the reasons behind the obtained result in the author's opinion. It is not enough to report other researcher opinions toward the obtained result but to lay down the reasons behind the obtained result using the actual output in the present study. Since it is the same for all subsections in the Discussion, please expand discussion with few more sentences and facts in each sub-section.

Conclusion

Well addressed and comply with the obtained results.

Author Response

Abstract

Point 1: Please state the number of animals in each group of grazing plus the ave. BW of the animals.

Response 1: The number of animals was 8 head, and the average BW was 21.86 kg. The informantion has been added in line 24.

Point 2: L25: please state what type of supplements are in the text.

Response 2: The supplements are concentrate and hay, which had been added to Abstract (line 25) and the “M & M” section (lines 85–86).

Introduction

Point 3: Please include a sentence (where appropriate) regarding the economic benefits of grazing that can add value to the indirect beneficial reasons behind doing this research.

Response 3: A sentence has been inserted in lines 51–52. The sentence is as follow: Reduced grazing duration requires less workforce, providing herdsmen more time to do other jobs.

M & M

Point 4: L84: How the ad libitum water was provided during grazing? In buckets?

Response 4: Yes, the drinking water was provided in buckets.

Point 5: What was the acre of grazing for each group?

Response 5: A total of 30 hectares of pasture were fenced off into three plots, the plots of 10 ha each were divided into four equal paddocks, giving a total of 12 paddocks of 2.5 ha each to avoid the possible bias on pasture availability between treatments. This was detailed in our previous study (Zhang et al., 2017).

Zhang, X.Q.; Jin Y.M.; Badgery, W.B.; Tana. Diet selection and n-3 polyunsaturated fatty acid deposition in lambs as affected by restricted time at pasture. Sci. Rep. 2017, 7, 15641.

Point 6: What happened during rain for housing the animals?

Response 6: Animals housed and fed concentrate and grass hay on a rainy day.

Point 7: How was the grazing management performed?

Response 7: All lambs began to access pasture at 6:00 h and were removed at 8:00 h, 10:00 h, 14:00 h and 18:00 h for 2H, 4H, 8H and 12H treatments, respectively. At the end of the time allowed at pasture for each treatment, lambs were separately housed in 32 individual pens and fed supplements of concentrate and grass hay during the rest of the day. This was detailed in our previous study (Zhang et al., 2014).

Zhang, X.Q.; Luo, H.L.; Hou, X.Y.; Badgery, W.B.; Zhang, Y.J.; Jiang, C. Effect of restricted time at pasture and indoor supple-mentation on ingestive behaviour, dry matter intake and weight gain of growing lambs. Livest. Sci. 2014, 167, 137-143.

Results

Point 8: Please be consistent in using decimal points in the value of the tables.

Response 8: All data in the tables , including Table S1, has been changed to two decimal points.

Discussion

Point 9: In each section, please improve the discussion by speculating the reasons behind the obtained result in the author's opinion. It is not enough to report other researcher opinions toward the obtained result but to lay down the reasons behind the obtained result using the actual output in the present study. Since it is the same for all subsections in the Discussion, please expand discussion with few more sentences and facts in each subsection.

Response 9: Thank you for yous useful comments. As your comments, we improve the discussion section.

Fitstly, we have added a short summery reason for gastrointestinal tract development in lines 241–243 (page 7). They are: In summary, shorter grazing durations are beneficial to the gastrointestinal tract development of lambs as these result in decreased foraging and more concentrated intake, which in turn result in higher nutrient digestibility.

Secondly, we have improved the expression in lines 279–283. The sentence is: On the other hand, the study by Zhang et al. [6] demonstrated that lambs that grazed for 2 or 4 h showed a remarkable decrease in extra energy expenditure through a significant reduction in their grazing activity; this, in turn, increased their available energy and resulted in a positive energy balance, thereby improving animal production. Overall, for all the reasons described above, grazing for shorter periods was beneficial to animal production.

Here reference of [28] was removed from text (line 282, page 8) and Reference list (after line 366, page 9).

Conclusion

Point 10: Well addressed and comply with the obtained results.

Response 10: Thank you for your positive comments.

Reviewer 3 Report

Dear authors,

The article is addressing an interesting topic of research. See below a few comments/suggestions to improve its quality after minor revision:

L65: Replace 'man-agement' by 'mana-gement'

L167: Replace 'Table 1' by 'Table 3'

L179: Replace 'Table 1' by 'Table 4'

L240: Replace 'sched-ule' by 'schedu-le'

Best regards,

Reviewer.

Author Response

Point 1: L65: Replace 'man-agement' by 'mana-gement'.

Response 1: That is auto linefeed. Now 'man-agement' is management in line 70.

Point 2: L167: Replace 'Table 1' by 'Table 3'.

Response 2: Correction has been made in line 177 (page 5).

Point 3: L179: Replace 'Table 1' by 'Table 4'.

Response 3: Correction has been made in line 190 (page 6).

Point 4: L240: Replace 'sched-ule' by 'schedu-le'.

Response 4: That is auto linefeed. Now 'sched-ule' is 'schedule' in line 256 (page 7).

Round 2

Reviewer 2 Report

The authors significantly improved the presentation of their work. No additional comments are required.